# Virological Passive Surveillance of Avian Influenza and Arboviruses in Wild Birds: A Two-Year Study (2023–2024) in Lombardy, Italy

**DOI:** 10.3390/microorganisms13050958

**Published:** 2025-04-22

**Authors:** Maria Cristina Rapi, Ana Maria Moreno Martin, Davide Lelli, Antonio Lavazza, Stefano Raimondi, Marco Farioli, Mario Chiari, Guido Grilli

**Affiliations:** 1Dipartimento di Medicina Veterinaria e Scienze Animali, Università degli Studi di Milano, Via dell’Università 6, 26900 Lodi, Italy; maria.rapi@unimi.it; 2Istituto Zooprofilattico Sperimentale della Lombardia e dell’Emilia Romagna “Bruno Ubertini” (IZSLER), Via Antonio Bianchi 7/9, 25124 Brescia, Italy; anamaria.morenomartin@izsler.it (A.M.M.M.); davide.lelli@izsler.it (D.L.); alavazza0@gmail.com (A.L.); 3Centro Recupero Fauna Selvatica (CRAS)—“Bosco WWF di Vanzago”, Via delle 3 Campane, 20043 Vanzago, Italy; stefanoraimondi67@gmail.com; 4S.C. Animali, Ambiente e One Health, Dipartimento Veterinario e Sicurezza degli Alimenti di Origine Animale, ATS Insubria, Via Ottorino Rossi 9, 21100 Varese, Italy; fariolim@ats-insubria.it; 5U.O. Veterinaria, Direzione Generale Welfare, Regione Lombardia, Piazza Città di Lombardia 1, 20124 Milano, Italy; mario_chiari@regione.lombardia.it

**Keywords:** avian influenza, West Nile virus, Usutu virus, H5N1, HPAIV clade 2.3.4.4b, wild birds, passive surveillance

## Abstract

Avian influenza (AI), caused by *Alphainfluenzavirus* (family *Orthomyxoviridae*), poses significant threats to poultry, biodiversity, and public health. AI outbreaks in poultry lead to severe economic losses, while highly pathogenic strains (HPAIVs) severely impact wild bird populations, with implications for biodiversity and potential zoonotic risks. Similarly, arboviruses such as West Nile virus (WNV) and Usutu virus (USUV) are emerging zoonoses. WNV can cause severe neurological diseases in birds, humans, and other animals, while USUV significantly affects blackbird populations and has zoonotic potential, though human cases remain rare. This study investigated avian viruses in 1654 wild birds from 75 species that died at the Wildlife Rescue Center in Vanzago, Lombardy, during 2023–2024. Necropsies were conducted, and virological analyses were performed to detect avian influenza viruses, WNV, and USUV. Among the tested birds, 15 were positive for H5N1 HPAIV clade 2.3.4.4b, all in 2023, including 13 *Chroicocephalus ridibundus*, one *Coturnix coturnix*, and one *Columba palumbus*. Additionally, 16 tested positive for WNV (15 for lineage 2 and one for lineage 1), one for USUV, and 11 co-infections WNV/USUV were recorded in 2023–2024. These findings underscore the importance of avian viral passive surveillance in identifying epidemiological trends and preventing transmission to other species, including mammals and humans.

## 1. Introduction

The need for a deeper understanding of the epidemiology of critical zoonoses, essential for safeguarding both human and animal health, alongside the development of increasingly sensitive diagnostic systems, has ushered in a new era in wildlife infectious disease surveillance. Over 70% of emerging zoonotic diseases originate from wildlife populations, emphasizing even more the importance of targeted monitoring efforts [1].

Among wildlife, avian species hold a central position in the ecology of numerous pathogens, acting as both reservoirs and vectors of microorganisms that have significant implications for public health and a high economic impact on the poultry production system [2]. Several viral diseases affect wild birds, some of which are of particular concern. Among these, avian influenza disease (AI) and vector-borne diseases such as those caused by West Nile virus (WNV) and Usutu virus (USUV) have drawn significant attention due to their zoonotic potential and economic impact [2,3,4].

Avian influenza, caused by viruses classified within the genus *Alphainfluenzavirus* of the family *Orthomyxoviridae* [5,6], primarily infects birds but exhibits a broad host spectrum. Viruses belonging to this genus are the most significant and widespread within the viral family. They are classified into subtypes based on the antigenic properties of their surface glycoproteins: haemagglutinin (HA) and neuraminidase (NA) [5,7,8]. To date, 18 HA and 11 NA subtypes have been identified. Of these, 16 HA and 9 NA subtypes are maintained in wild bird populations, contributing to the natural circulation of avian influenza viruses (AIVs) [5,8,9,10]. These viruses are further categorized into low pathogenicity (LPAIV) and high pathogenic (HPAIV) strains based on the severity of clinical signs observed in birds, especially in poultry, with HPAIV responsible for lethal systemic infections [3,11,12,13]. While HPAI viruses infections have been considered as rarely occurring in wild birds, LPAIV strains have been isolated in at least 105 wild bird species across 26 families [5,9], with species within the orders *Anseriformes* (mainly ducks, geese, and swans) and *Charadriiformes* (gulls, terns, and shorebirds) officially recognized as reservoirs of LPAIVs [9,11,14,15]. Moreover, given their migratory behavior, birds belonging to these orders are considered principal contributors to the global dissemination of AIVs and key vectors in transmitting these viruses to poultry and mammalian hosts [3,9,14,16]. Considering that detecting H5/H7 HPAI in poultry entails adopting strict veterinary police measures, with severe economic losses, it is crucial to ensure the monitoring of AI [17,18]. Additionally, the identification of HPAI viruses with genetic markers indicating potential adaptation for replication in mammals underscores the importance of systematic surveillance systems [5,19,20].

Other avian viruses for which monitoring wild birds becomes essential for protecting human and animal health are the West Nile virus and Usutu virus. Considered emerging zoonoses in Europe and Italy, where they have now been endemic for more than a decade, the West Nile disease (WND) and Usutu disease are caused by two distinct but related neurotropic flaviviruses (genus *Flavivirus*; family *Flaviviridae*). These viruses are maintained in the environment through an enzootic transmission cycle involving wild birds, recognized as the reservoir hosts, and mosquitoes, in particular *Culex* sp., as intermediate vectors [21,22,23]. While numerous bird species are susceptible to WNV and USUV, members of the family *Passeriformes* (especially house sparrows, corvids, and blackbirds) are identified as the most significant avian reservoir in the epidemiology of WNV and Usutu virus in Europe and the Americas [24,25]. Humans and other mammals, including horses, can be infected through bites of infected mosquitoes. Still, they are considered spillovers and “dead-end” hosts, as they do not contribute to further virus transmission [21,25,26]. Despite this, WNV is particularly concerning due to its potential to cause lethal neurological forms in a low percentage of infected “dead-end” hosts. Conversely, USUV appears to be more pathogenic in some wild bird species but rarely causes severe disease in humans, typically resulting in asymptomatic infections [21,27,28]. However, growing evidence suggests that the public health risk associated with USUV circulation should be reconsidered [29,30]. Since no effective therapies or vaccines exist for treating and preventing these arboviruses in humans, establishing a rigorous monitoring system becomes essential for preventing outbreaks [31,32]. In particular, since many infections in humans could be asymptomatic, there is a need to avoid iatrogenic infections through blood and organ donations [33,34,35,36].

Given the ecological roles of wild birds in the distribution and persistence of AIVs, WNV, and USUV, systematic surveillance of avian populations is essential. Monitoring wild birds offers valuable insights into pathogen transmission and virus circulation, enabling the early detection of outbreaks in food-producing animals and infection in companion animals and humans [37]. In light of these considerations, national surveillance plans for AI [38] and arbovirosis [39] are regularly performed in Italy.

In alignment with the Avian Influenza National Surveillance Plan and the National Plan for the Prevention, Surveillance, and Response to Arbovirosis (PNA), the present study aimed to conduct a post-mortem investigation of AIVs, WNV, and USUV in wild birds deceased at a wildlife rescue center in Lombardy during 2023–2024. Given the extensive regional coverage of wetland areas, Lombardy hosts a significant number of resident and overwintering aquatic birds. In light of this, and considering the high density of poultry farms in the region, monitoring wild bird populations holds strategic importance for safeguarding both public health and the poultry production system.

## 2. Materials and Methods

### 2.1. Animals

The study was carried out from 2023 to 2024. During this time, wild birds belonging to different species were monitored and subjected to subsequent virological control to detect AIVs, WNV, and USUV. All wild birds considered died spontaneously or were humanely euthanized at the Wildlife Rescue Center WWF of Vanzago (Lombardy region), a facility dedicated to the sheltering and recovery of wild animals. The euthanasia protocol was applied by the center’s veterinary medical director in accordance with animal welfare regulations and legal requirements, in cases of severe illness, injury, or debilitation, or when release into the wild was not feasible. Therefore, no animals were deliberately or specifically sacrificed for the purposes of this study.

Birds were sampled consistently throughout the two-year period. Seasonal variation in sample numbers reflects natural fluctuations in bird admissions associated with breeding, the nestling season and migration periods. A graphical representation of the number of samples collected per month in 2023 and 2024 is available in the Appendix A.

At the Rescue Center of Vanzago, all wild animals undergo a medical assessment upon admission. Each specimen is assigned a unique identifier, and a corresponding clinical record is created. This record provides veterinarians and staff with detailed information on the individual, including clinical data, anamnesis, and the date and location of discovery.

### 2.2. Samples Collection

Upon death, all wild birds were frozen at −20 °C and transported weekly to the Department of Veterinary Medicine and Animal Sciences at the University of Milan, Lombardy, Italy. Here, each specimen underwent a necropsy exam to collect tissue samples, including the brain, trachea, lungs, heart, spleen, kidneys, and intestine. All organs collected from each specimen were preserved in sealed Petri dishes (one dish per animal), labeled with the species, and identified with the unique identification code assigned upon the animal’s admission to the rescue center. The samples were then frozen at −20 °C and stored for subsequent investigation for the presence of AIVs, WNV, and USUV.

A dedicated document labeled W02, titled “Surveillance of Resident Birds from Synanthropic Species—Monitoring Mortality in Wild Birds”, was filled for every tissue sample. This document was specifically designed by National Veterinary Authorities to ensure accurate tracking and handling of samples throughout the collection and testing process as part of the West Nile disease surveillance program.

### 2.3. Detection of Viral Genomes

The collected samples were sent to the Virology Department of Istituto Zooprofilattico Sperimentale della Lombardia e dell’Emilia Romagna (IZSLER), Brescia, where they were analyzed using molecular techniques.

Following mechanical homogenization of the organs collected from each individual, viral RNA was extracted from each sample using the QiaSymphony SP/AS (Qiagen, Hilden, Germany) with the DSP Virus/Pathogen mini kit (Qiagen, Hilden, Germany) according to the manufacturer’s protocol, with a final elution volume of 60 μL. The AIV genome detection was performed using a real-time RT-PCR whose target is the M-gene, as described in [40]. Positive samples were subsequently analyzed using two real-time RT-PCR assays for the detection of the H5 and H7 genomes [40,41]. Samples positive for H5 or H7 were sent to the National Reference Laboratory for Avian influenza and Newcastle disease at the Istituto Zooprofilattico Sperimentale delle Venezie (IZSVe) in Legnaro (Padua) for confirmation of positivity and for genomic typing. Laboratory methods were performed in accordance with internal procedures [42]. At this facility, neuraminidase subtyping of avian influenza-positive samples was performed using real-time RT-PCR with multiple oligonucleotide sets (protocol code PDPVIR1004) [43,44,45]. H5 pathotyping and clade definition were conducted using real-time RT-PCR assays targeting the multi-basic cleavage site specific to highly pathogenic H5 strains (protocol codes PDPVIR1005 and PDPVIR125) [46].

Viral RNAs extracted from bird samples were further analyzed at the IZSLER for the detection of WNV and USUV genome by two real-time RT-PCR assays [47,48]. WNV positive samples from the first real-time RT-PCR were further tested using additional real-time RT-PCR to identify WNV lineages L1 and L2 [49]. Samples positive for WNV and USUV were sent to the National Reference Centre for Exotic Diseases (CESME) of the Istituto Zooprofilattico Sperimentale dell’Abruzzo e del Molise (IZSAM) in Teramo for confirmation of positivity.

## 3. Results

During the two-year period, 1654 wild birds from 75 different species belonging to 19 orders were monitored for AIVs, WNV, and USUV. Of these, 815 (49.3%) specimens were collected in 2023 and 839 (50.7%) were collected in 2024 (for more details see Appendix A.

The majority of the specimens analyzed in this two-year period belonged to the orders *Passeriformes* (*n* = 805, 49%), *Columbiformes* (*n* = 260, 16%), *Apodiformes* (*n* = 133, 8%), and *Anseriformes* (*n* = 102, 6%), collectively accounting for approximately 79% of all individuals examined in the study (Figure 1a). The remaining 21% consisted of 15 other orders, with the orders *Caprimulgiformes*, *Podicipediformes*, *Bucerotiformes*, *Ciconiiformes*, *Suliformes*, *Cuculiformes*, and *Gruiformes* as the most underrepresented in the sample (Figure 1b).

The Wildlife Rescue Center of Vanzago is the largest in northern Italy in terms of the number of annual hospitalizations, serving as a reference facility for other Italian regions. It usually admits animals from more than half of the provinces of Lombardy and covers almost 70% of the regional territory. The birds that died in the rescue center included in this study came from different provinces, among which the most representative were Milan (*n* = 1092, 66%), Varese (*n* = 212, 13%), and Monza Brianza (*n* = 174, 10%) (Figure 2a). The wild birds from these provinces constituted 89% of the total population sampled for the present study. The remaining 11% (*n* = 176) consisted of specimens originating from 12 other provinces inside, i.e., Como (*n* = 93), Lodi (*n* = 49), Pavia (*n* = 13), Bergamo (*n* = 7), Lecco (*n* = 2), Cremona (*n* = 2), Brescia (*n* = 1), and outside Lombardy, i.e., Novara (*n* = 4), Verbania (*n* = 2), Macerata (*n* = 1), Lucca (*n* = 1), and Genova (*n* = 1) (Figure 2b).

### 3.1. Prevalence of AIVs

In total, 15 out of the 1654 samples tested for AIVs in this two-year period tested positive for *Alphainfluenzavirus*. All positive samples were identified as subtype H5N1 HPAIV, clade 2.3.4.4b. These positive cases were recorded exclusively in 2023, with an overall prevalence of 1.8% (*n* = 15/815) for that year. All positive cases were recorded in February and March 2023 in the Lombardy provinces of Milan, Varese, and Monza Brianza (Figure 3).

Among the positive specimens, 13 were identified as *Chroicocephalus ridibundus*, one as *Coturnix coturnix*, and one as *Columba palumbus*, as summarized in Table 1.

Considering the taxonomic orders of these specimens, AI prevalence was defined to be 46% (*n* = 13/28) and 33% (*n* = 1/4) in the *Charadriiformes* order and in the *Galliformes* order, respectively. Within the *Columbiformes* order, only one positive case was detected among 118 specimens, resulting in a prevalence of 0.8%.

### 3.2. Prevalence of WNV and USUV

In total, 28 out of the 1654 samples tested in 2023–2024 for WNV and USUV tested positive for these arthropod-borne viruses, with an overall prevalence of 1.7% (*n* = 28/1654).

Passive surveillance for arboviruses performed in 2023 revealed 26 positive tissue samples out of 815 specimens analyzed, originating from the Lombardy provinces of Milan, Varese, Como, and Monza Brianza (Figure 4). Of these, 13 samples tested positive for WNV lineage 2 (L2), one for WNV lineage 1 (L1), and one for USUV. Additionally, 11 cases of co-infection with WNV L2 and USUV were detected (Figure 4).

Table 2 summarizes the arboviruses-positive results recorded in 2023. All these positive cases were detected between July and September 2023.

During the year 2023, the prevalence of WNV (including both L2 and L1) in the sampled population was 1.7% (*n* = 14/815), while the prevalence of USUV was 0.1% (*n* = 1/815). Co-infection with WNV L2 and USUV had a prevalence of 1.3% (*n* = 11/815).

When considering WNV, given the 14 positivities recorded, the highest number of positive cases was observed in *Passeriformes* (*n* = 7/14), with the *Corvidae* family being the most represented among the positive specimens, followed by *Accipitriformes* (*n* = 4/14), *Falconiformes* (*n* = 2/14), and *Apodiformes* (*n* = 1/17). The prevalence of WNV in the order *Accipitriformes* and *Falconiformes* was 28.6% (*n* = 4/14) and 7.4% (*n* = 2/27), respectively, followed by *Passeriformes* with a prevalence of 1.8% (*n* = 7/397) and *Apodiformes* with a prevalence of 1.6% (*n* = 1/63). The WNV L1 genome was identified in a single specimen of *Falco tinnunculus* in August 2023, while all other positive cases involved WNV L2.

During the same year, USUV was detected in only one specimen of *Turdus merula*, resulting in a prevalence of 0.2% (*n* = 1/397) within the order *Passeriformes*.

Co-infection cases with WNV L2 and USUV were predominantly observed in the *Passeriformes* order (*n* = 7/11), with three cases detected in *Turdus merula*, one in *Pica pica*, and three in *Corvus cornix*. Co-infection prevalence within *Passeriformes* was 1.8% (*n* = 7/397). Additional co-infection cases were recorded in one specimen each of *Accipiter gentilis*, *Falco tinnunculus*, *Columba palumbus*, and *Tachymarptis melba*. The prevalence of co-infection in these orders was 7% (*n* = 1/14), 3.7% (*n* = 1/27), 0.8% (*n* = 1/118), and 1.6% (*n* = 1/63), respectively.

Out of 28 positivities for arboviruses detected in this two-year study, the remaining two were identified in September 2024 in wild birds that came from the Lombardy province of Milan (see Figure 4). Both resulted positive for WNV L2, with a total of 15 WNV L2-positive specimens in 2023–2024. The overall prevalence of WNV in the sampled population in 2024 was 0.2% (*n* = 2/836). More in detail, one positivity was recorded in a *Corvus cornix* specimen and one in a *Garrulus glandarius* specimen. Given this, in 2024, the prevalence of WNV in *Passeriformes* order was 0.5% (*n* = 2/408).

## 4. Discussion

Data obtained from the passive surveillance of wild birds provided evidence of the circulation of avian influenza virus, West Nile virus, and Usutu virus in free-living avian species in the Lombardy region during 2023–2024.

With reference to the AI, molecular analysis identified the type H5N1 HPAIV, clade 2.3.4.4b, as the responsible agent for all of the 15 positive samples recorded in February and March 2023. During the early months of 2023, this AIV was confirmed as the causative agent of numerous outbreaks affecting both domestic and wild birds across several European countries [50]. In Italy, two HPAI virus outbreaks were reported in poultry, along with significant circulation of the virus among colony-breeding seabird species and waterfowl [50]. Genetic analyses demonstrated that the virus persisted within resident free-living avian species throughout the summer. Consequently, the 2022–2023 epidemiological year did not exhibit a distinct onset of the HPAI epidemic season, with positive cases being detected even during a traditionally “ low-risk “ period for AIV circulation [50]. In this two-year study, the majority of the positive samples were found in *Chroicocephalus ridibundus*, aligning with other Italian and European reports, documenting significant H5N1 HPAI outbreaks and associated die-offs in seabirds during the same period [50,51,52]. Northern Italy, in particular, experienced an unprecedented rise in black-headed gull mortality due to H5N1 HPAIV, clade 2.3.4.4b, particularly in wetland areas [50,51,53]. Given the large resident gull populations in Italy [54,55,56], this scenario resulted in an increased risk of influenza virus transmission to poultry, especially during spring when younger birds—known for higher pathogen shedding—are more prevalent [51].

In 2024, the number of H5N1 HPAIV detections in both wild and domestic birds was significantly lower compared to the same period in 2023 [50,57]. This decline may be attributed, among other factors, to increased immunity within previously affected wild bird populations, resulting in decreased transmission rates. This condition has led to fewer mortality events in wild birds across Europe since the beginning of the 2023–2024 epidemiological year and a reduced number of outbreaks in poultry compared to the same period in the previous year [57]. In light of this, data collected in the present study confirmed the important role that members of the order *Charadriiformes* had in the spread of H5N1 HPAIV, clade 2.3.4.4b, in 2023 in Italy.

The same HPAIV subtype has also been detected in outbreaks involving various mammalian species worldwide [4,52]. Although the virus maintains a preferential binding affinity for avian-like receptors, several mutations linked to an increased zoonotic potential were identified [50,58].

Given the significant role of wild birds in transmitting influenza viruses to poultry and the potential human health threat [59], the necessity of integrated surveillance plans becomes evident. Such plans require collaboration among various stakeholders, with wildlife rescue centers playing a crucial role. These centers not only contribute to the conservation of native fauna but also serve as epidemiological observatories for monitoring wild animals’ health. In this study, the involvement of a wildlife rescue center enabled the investigation of species that are typically challenging to access, such as birds of prey. Moreover, this collaboration facilitated the extension of the epidemiological survey to include species that were not listed in the official European surveillance list, valid until 2023 [60]. Until the recent update [61], the European Union’s surveillance list comprised fifty wild bird species that were routinely monitored for H5 HPAI viruses. This list provides a framework for operators to prioritize certain bird species for surveillance, thereby enhancing the early warning system for AI [60,61]. In the present study, of the 67 species monitored in 2023, only 10 were included in the European list in force at that time [60]. With these exceptions, virological surveillance would not have been feasible for the other species without the collaboration of the wildlife rescue center in Vanzago.

Positivity for H5N1 HPAIV, clade 2.3.4.4b, was also detected in a European Quail (*Coturnix coturnix*) and a Common Woodpigeon (*Columba palumbus*). The European Quail is well-documented as being susceptible to AIVs [62,63,64]. On the other hand, the role of *Columbiformes* in the maintenance of AIVs remains less understood. Literature on natural infections in Columbiformes is scarce, and these birds have demonstrated low susceptibility to AIVs in experimental settings [65,66,67]. Given the dynamic nature of HPAIV infection, *Columbiformes* have been included in the recently updated European list of prioritized species for effective AI surveillance [61] despite sporadic reports during the last two years. This inclusion underscores the need for continued monitoring of this disease for human and animal health protection.

For WNV and USUV, this study highlighted the predominance of WNV cases, with 16 samples testing positive compared to only one case of USUV during the two-year period. WNV is one of the most widespread mosquito-borne flaviviruses globally [26,68]. First isolated in Uganda in 1937, in August 1999, it was identified in the United States where, in the New York City area, it caused deaths in flocks of American Crows (*Corvus brachyrhynchos*), as well as neurological forms in horses and people [69]. In Europe, WNV is currently endemic in central and southeastern Europe [68], with circulating strains belonging to L1 and L2, which are recognized as the most virulent [21]. In Italy, since 2014, there has been a progressive increase in the circulation of WNV until 2018, an exceptional year in terms of the number of cases of infection and disease in humans, equids, and reservoir species [37,70]. In 2023, a total of 332 confirmed cases of WNV infection in humans were reported in Italy, with the majority occurring in the Lombardy region [71]. By comparison, the number of confirmed WNV cases in humans increased in the 2024 vector season, but Lombardy was not the most affected region [72].

The lesser-known USUV was first observed in Europe in 2001 [73], even though this virus has probably been present in Europe since 1996. Subsequently, USUV spread throughout Europe, causing mortality in bird populations and, occasionally, severe human neurological cases [32,74].

Concerning the reservoir hosts of WNV and USUV, several studies recognize the orders *Passeriformes* (especially the families *Corvidae*, *Fringillidae,* and *Passeridae*), *Charadriiformes* (family *Laridae*), *Accipitriformes*, *Falconiformes*, and *Strigiformes* as those comprising the species most susceptible to these viruses [75,76,77]. In the present study, among the WNV-positive cases, several species were identified as particularly susceptible, including *Corvus cornix*, *Pica pica*, *Carduelis carduelis*, *Accipiter gentilis*, *Falco tinnunculus*, and *Turdus merula*. The latter resulted particularly susceptible also to USUV infection [74], with the only recognized case of positivity for this virus.

Most detected WNV cases involved L2, with the exception of a specimen of *Falco tinnunculus*, which tested positive for WNV L1. WNV is currently classified into eight distinct phylogenetic lineages, with L1 and L2 associated with severe clinical manifestations in humans and horses [78]. Prior to 2011, WNV circulation in Italy was predominantly attributed to L1 strains [79]. However, in 2011, the introduction of Eastern European L2 was documented in northeastern Italy. This lineage rapidly disseminated across the country, ultimately supplanting the previously circulating L1 strains [79,80]. Following a decade-long absence, WNV L1 re-emerged in areas surrounding the Po River delta, raising questions about whether it represented a reintroduction or the reoccurrence of strains that had persisted undetected [79]. This reappearance has further complicated the epidemiological landscape, as both L1 and L2 now co-circulate in several regions, increasing the potential for overlapping outbreaks [71,72] and highlighting the importance of rigorous WNV surveillance systems across Italy.

Although the results of this study confirm positivity in avian orders characterized by species most susceptible to WNV and USUV infection, the high number of cases in scavenging and predatory species is notable. Although the primary transmission cycle of WNV and USUV involves the ‘mosquito–bird–mosquito’ pathway, studies have shown that some bird species can also become infected through the ingestion of viremic prey [75,81]. The significance of oral transmission likely differs among raptor species, influenced by different factors such as the susceptibility of their prey to WNV. Despite this, various studies suggest that raptors are among the avian species most frequently affected during WNV outbreaks [82].

The fact that 14 out of 16 positives for WNV were recognized in *Turdus merola*, *Pica pica*, and *Corvus cornix* is also in line with the literature. The documented feeding preference of *Culex* mosquitoes for the Common Blackbird (*Turdus merula*) has significant epidemiological implications. A decrease in the *Turdus merula* population due to migration or extensive anthropogenic changes in the environment has been associated with an increased incidence of *Culex* blood meals on corvids [83]. Consequently, corvid species play a critical role in the epidemiology of WNV in Europe. In contrast, the other species identified as WNV-positive in this study are infrequently implicated in disease outbreaks. This observation is particularly relevant when considering the Common Woodpigeon (*Columba palumbus*). Species within the order *Columbiformes*, as well as those belonging to *Pelecaniformes*, *Psittaciformes*, and *Galliformes*, do not typically develop viremic levels sufficient to support the transmission of the pathogen and, consequently, the ongoing virus circulation in the environment [75].

The detection of WNV in one specimen of Alpine Swifts (*Tachymarptis melba*) is also interesting, given the fact that this species is recognized as potentially involved in the introduction, amplification, and spread of the pathogen from southern Africa to Europe [84].

WNV and USUV co-circulate in Europe, sharing both vector hosts and some reservoir hosts. The overlap of these arboviruses increases the likelihood of co-infection cases, which in susceptible species have already been reported, although not as frequently, due to antibody cross-reactivity [21,27,36,85,86]. Considering this, the finding of 11 co-infections with WNV L2 and USUV within a short period (July–September 2023) is significant. It highlights the need for more in-depth studies on this topic. Indeed, co-infections may present a different outcome than mono-infections, thus having a different weight regarding public health protection [21,86].

Lombardy hosts significant numbers of overwintering aquatic birds, and many of its wetlands are of conservation importance, with a high degree of regional coverage. Regarding surface area, 85% of the wetlands surveyed in Lombardy consist of natural lakes, making it the Italian region most abundant in lakes, followed by rivers, which account for 10.6%. Other types of wetlands, such as marshy areas, collectively cover approximately 5% of the total wetland surface [87,88]. The abundance of water in the region serves as a natural attractant for many species of wild aquatic birds, which use these habitats as important stopover sites during migration. Over the two years considered in this study, the *International Waterbird Census* (IWC) conducted in Lombardy recorded over 120,000 overwintering aquatic birds from various species [55,56]. Among these, *Anatidae* and *Laridae* species populations are particularly well-represented, with the mallard (*Anas platyrhynchos*) and the black-headed gull (*Chroicocephalus ridibundus*) being some of the most abundant species [55,56]. These two species are key reservoirs of avian influenza viruses.

Furthermore, these wetlands, in particular marshy areas, meet the criteria for a successful transmission cycle of West Nile virus and Usutu virus, as they are rich in mosquitoes, often situated near equine and human populations, and, as previously mentioned, attract large numbers of migratory birds [89]. Considering this, in the PNA, Lombardy is classified as a high-risk region for the circulation of WNV, where infection episodes have been repeatedly observed over the years [39].

In light of these aspects, and given that the majority of poultry farms are located in the northern regions of Italy—with Lombardy accounting for 15% of national poultry farms and 11% of the total poultry population—the monitoring of wild bird populations in Lombardy is of paramount importance [90]. Such surveillance allows for the rapid identification of the presence and circulation of avian viruses, thereby safeguarding both the poultry sector and public health.

## 5. Conclusions

The observation of avian influenza outbreaks in wild birds and poultry is no longer an uncommon phenomenon. Outbreaks in avian species have been established with increasing frequency over the past 15 years, with at least ten H5 HPAI epidemic events in Europe resulting in mass mortality events among poultry and free-living birds during this period [91]. While prior to 2009, AI outbreaks were predominantly associated with the H5N1 subtype, clade 2.2, the period from 2014 onward has seen a shift, with most European outbreaks now driven by HPAI H5N1 clade 2.3.4.4b viruses. This lineage, which has become predominant globally, is characterized by high pathogenicity and the presence of genetic markers indicating a likely adaptation to mammals [4,50,51,52,92].

In parallel, Europe and Italy have witnessed an increase in the incidence of WNV and USUV outbreaks in both wild birds and humans since 2020. Notably, in 2023, 709 human cases of WNV were reported across nine EU countries, with 332 cases occurring in Italy [71]. Although this represents a decrease from the numbers reported in 2022 and 2018, the broader geographical distribution of cases indicates the introduction and establishment of the virus in new areas [70,93]. This expanding range, coupled with the increasing incidence and severity of symptoms, underscores the need to view WNV and USUV as serious emerging threats to public health [27].

To prevent outbreaks of AI, WND, and Usutu disease in livestock, pets, and humans, the establishment of a comprehensive surveillance system targeting the bird population is imperative. Such a system must incorporate both active and passive surveillance measures. Monitoring wild birds plays a crucial role in this context, as it provides an early warning system for potential or actual threats to reservoir populations [94,95,96,97].

While passive surveillance, as employed in the present study, has certain limitations—such as procedural delays and potential sample heterogeneity due to variations in avian species distribution [94,98]—the data generated are invaluable, and the findings of this study demonstrate the utility of passive surveillance in identifying and assessing emerging threats. However, integrating passive measures with those implemented in active surveillance is essential for a more comprehensive epidemiological understanding of AI, WND, and Usutu disease, as also reported by Trogu et al. [53]. In Lombardy, the integration of passive and active surveillance in wild birds is already in place, as both approaches are mandated at the national level under the National Plan for Avian Influenza Surveillance [38]. This comprehensive surveillance framework constitutes a highly effective early warning system, which is crucial not only for the protection of public health but also for safeguarding the poultry industry. Moreover, the PNA 2020–2025 [39] further strengthens this approach by incorporating entomological, human, and animal surveillance, with a specific focus on wild birds as sentinels for arboviral circulation.

By ensuring the timely detection and assessment of these emerging threats, this integrated strategy plays a fundamental role in mitigating potential zoonotic risks to both human and animal populations [96,97].

## Figures and Tables

**Figure 1 microorganisms-13-00958-f001:**
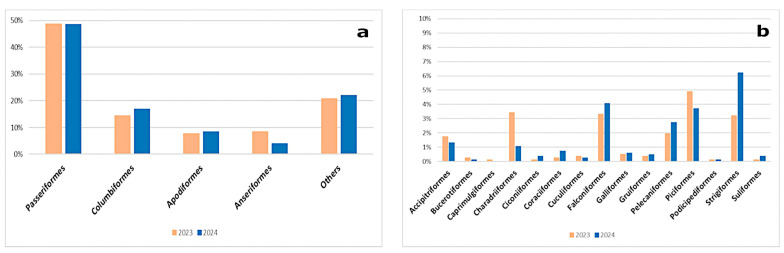
Passive surveillance in wild birds. (**a**) Proportional distribution of collected wild birds during 2023–2024, categorized by the most represented taxonomic orders; (**b**) Proportional distribution of wild birds categorized by less represented taxonomic orders (“Others” in (**a**)).

**Figure 2 microorganisms-13-00958-f002:**
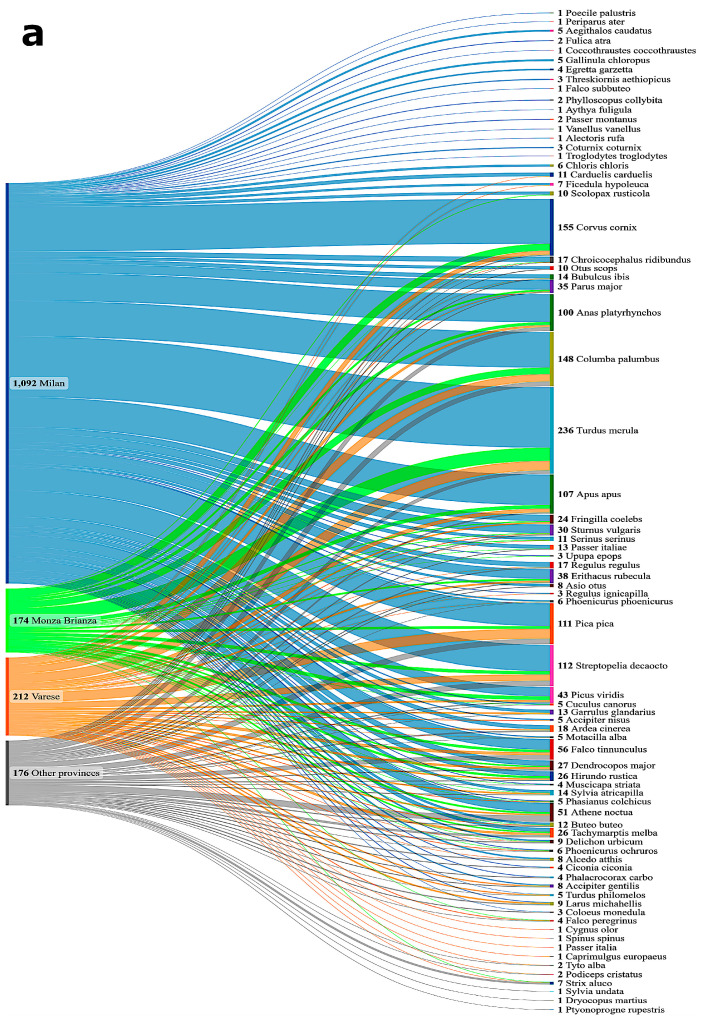
Provinces of origin of wild birds analyzed. (**a**) Provinces within Lombardy region contributing to wild bird samples analyzed during 2023–2024, showing the most representative areas. (**b**) Geographic origins of wild birds analyzed during 2023–2024, including provinces from both Lombardy and neighboring Italian regions (“Other provinces” in (**a**))”.

**Figure 3 microorganisms-13-00958-f003:**
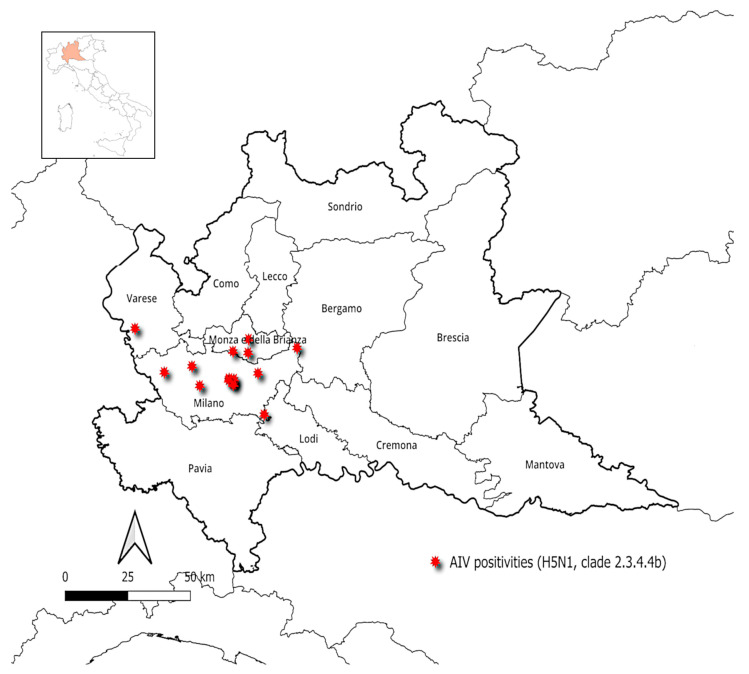
Locations of avian influenza virus (H5N1 HPAIV clade 2.3.4.4b) positive cases in Lombardy, detected in wild birds during February and March 2023 (QGIS 3.34.3 map).

**Figure 4 microorganisms-13-00958-f004:**
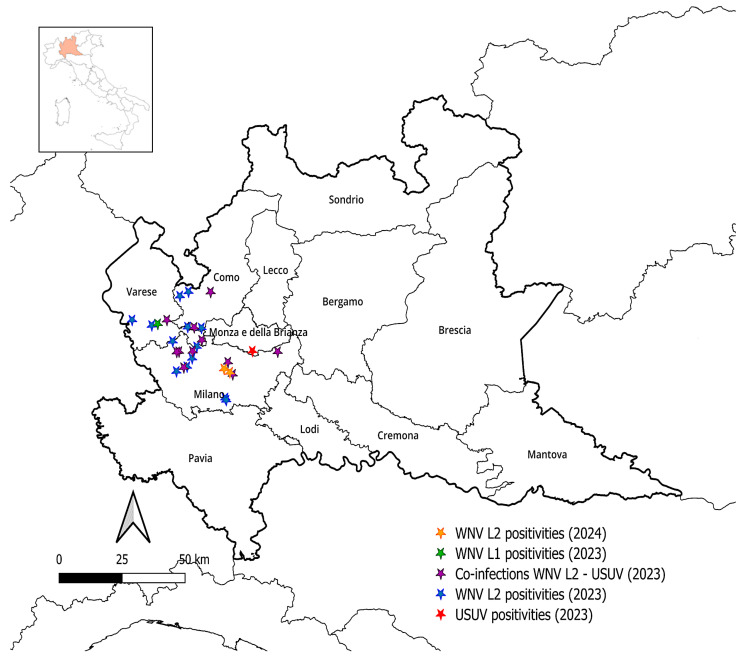
Geographic distribution of West Nile virus- and Usutu virus-positive cases in wild birds sampled in Lombardy during 2023–2024 (QGIS 3.34.3 map).

**Table 1 microorganisms-13-00958-t001:** Summary of avian influenza-positive cases (H5N1 HPAIV) by order and species detected in 2023.

Order	Species	Total (2023)	Positivities for AIV	Type
*Charadriiformes*	Black-headed Gull *(Chroicocephalus ridibundus)*	17	13 pos	H5N1 HPAIV, clade 2.3.4.4b
*Galliformes*	Common Quail *(Coturnix coturnix)*	1	1 pos	H5N1 HPAIV, clade 2.3.4.4b
*Columbiformes*	Common Woodpigeon *(Columba palumbus)*	69	1 pos	H5N1 HPAIV, clade 2.3.4.4b
		Total	15	

**Table 2 microorganisms-13-00958-t002:** Summary of WNV and USUV positive cases in wild birds, including co-infections, recorded in 2023.

Order	Species	Total (2023)	Positivities for WNV Only	Positivities for USUV Only	Co-Infection Condition
*Accipitriformes*	Northern Goshawk *(Accipiter gentilis)*	7	4 pos WNV-L2	-	-
	Eurasian Buzzard *(Buteo buteo)*	4	-	-	1 pos WNV-L2 and USUV
*Apodiformes*	Alpine Swift *(Tachymarptis melba)*	9	1 pos WNV-L2	-	1 pos WNV-L2 and USUV
*Columbiformes*	Common Woodpigeon *(Columba palumbus)*	69	-	-	1 pos WNV-L2 and USUV
*Falconiformes*	Common Kestrel *(Falco tinnunculus)*	25	1 pos WNV-L2 1 pos WNV-L1	-	1 pos WNV-L2 and USUV
*Passeriformes*	European Goldfinch *(Carduelis carduelis)*	6	1 pos WNV-L2	-	-
	Hooded Crow *(Corvus cornix)*	72	4 pos WNV-L2	-	3 pos WNV-L2 and USUV
	Eurasian Magpie *(Pica pica)*	51	2 pos WNV-L2	-	1 pos WNV-L2 and USUV
	Common Blackbird *(Turdus merula)*	130	-	1 pos	3 pos WNV-L2 and USUV
		Total	14	1	11

## Data Availability

The original contributions presented in the study are included in the article; further inquiries can be directed to the corresponding author.

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
