# Peer review of "Virological Passive Surveillance of Avian Influenza and Arboviruses in Wild Birds: A Two-Year Study (2023–2024) in Lombardy, Italy"

_microorganisms, 2025, doi:10.3390/microorganisms13050958_

Round 1
Reviewer 1 Report
Comments and Suggestions for Authors
To Editor, Authors
The manuscript “Virological Passive Surveillance of Avian Influenza and Arboviruses in Wild Birds: A Two-Year Study (2023-2024) in Lombardy, Italy”, Microorganisms Journal is of interest for virologists, especially for veterinary virology researchers: for epidemiologists of Avian Influenza and some of vector-born diseases among wild birds; for veterinarian specialists.
The authors describe the passive surveillance in dead avian species and showed the data generated are invaluable, and the findings of this study demonstrate the utility of passive surveillance in identifying and assessing emerging threats. The discussion section seems to me very interesting and meaningful.
But still has some questions to be addressed.
General Comments:
- Since you tested humanely euthanized birds, please provide the details of Bioethical committee
- Lines 152-153 During the two-year sampling, did you observe the seasonality, specific months? Were there peaks in the number of samples in different months, or was everything uniform?
- The main question - how did you identify Clade 2.3.4.4.b? for example Line 252. To do this, you need to sequence the HA segment. There is no information in the methods.
Specific Comments:
- Lines 19, 51 – please check the taxonomy of Influenza virus according to the ICTV
- Line 138 I am not sure clearly what do you mean: “homogenized organ pools of each sample“
- Line 134 Possibly the subsection title could be changed for “PCR detection” (for example) as you did not perform the Molecular analysis
Author Response
|
Response to Reviewer 1 Comments
|
|
We thank the reviewer for the time dedicated to evaluating our manuscript and for the positive and encouraging comments. Please find the detailed responses below and the corresponding revisions in track changes in the re-submitted files.
|
|
General comments |
|
Comment 1: Since you tested humanely euthanized birds, please provide the details of Bioethical committee.
|
|
Response 1: Thank you very much for your comment. We apologize for the misunderstanding. Indeed, the manuscript does not clearly state that the wild birds included in this study were all admitted to the Wildlife Rescue Center of Vanzago and had either died naturally or had been humanely euthanized by the center’s veterinary medical director, in accordance with animal welfare regulations and legal requirements, as they were severely ill, injured, debilitated, or unable to be released back into the wild. Therefore, no animals were deliberately or specifically sacrificed for this research. All animals had either died naturally or were euthanized for reasons unrelated to the study, solely due to their health conditions. To clarify and emphasize this important point, we have added a specific statement in the revised manuscript (line 116-120).
|
|
Comment 2: Lines 152-153 During the two-year sampling, did you observe the seasonality, specific months? Were there peaks in the number of samples in different months, or was everything uniform?
|
|
Response 2: Thank you for your comment. In this study, wild birds that died at the Wildlife Rescue Center were sampled and subsequently analyzed each month throughout both years (2023–2024). This approach, which did not rely on the typical seasonality of the diseases investigated, allowed us to detect AIV positivity during a period generally considered to be low-risk for viral circulation (February–March 2023). Regarding the second part of your comment, despite our decision to sample individuals year-round, seasonal numerical variations in the number of deceased and sampled animals do occur. Specifically, the number of analyzed samples begins to increase in spring (March–April), peaks between May and August, and then decreases in autumn, reaching its lowest levels during winter (December–February). This trend, observed consistently in both 2023 and 2024, reflects bird biology and, in particular, the migratory behavior of certain species. The spring–summer period corresponds to breeding and migration seasons, resulting in a greater number of admissions to the center, whereas the autumn–winter period is characterized by fewer admissions. Following your suggestion, we have added a brief explanation of this aspect to the manuscript (lines 121-123). Additionally, one graph showing the monthly distribution of samples collected in 2023 and 2024 have been included in Supplementary File 1 (Figure S1), added at the article after your revision.
|
|
Comment 3: The main question - how did you identify Clade 2.3.4.4.b? for example Line 252. To do this, you need to sequence the HA segment. There is no information in the methods.
|
|
Response 3: Thank you for pointing this out. It is important to note that, in Italy, all samples testing positive for avian influenza viruses H5 and H7 must be sent, according to national legislation and following the indications of the Ministry of Health, to the National Reference Laboratory located at the IZSVe in Padua. At this institution, H5 and H7 influenza viruses undergo specific analyses (Reverse Transcriptase real-time PCR essays) to determine their pathotype and their clade, as described in protocols PDPVIR1005 and PDPVIR125, as well as neuraminidase subtyping (protocol code PDPVIR1004). These procedures are routinely followed by genotypic and phylogenetic analyses, conducted after whole-genome sequencing of the viral genome using Next-Generation Sequencing (NGS) technology, but we did not consider this information in our research. To clarify these steps, we have added a paragraph to the “Materials and Methods” section (subsection Detection of Viral Genomes, lines 162–167), and included bibliographic references.
|
|
Specific comments |
|
Comment 1: Lines 19, 51 – please check the taxonomy of Influenza virus according to the ICTV
|
|
Response 1: Thank you for your comment. We have checked the ICTV and consequently modified the text.
|
|
Comment 2: Line 138 I am not sure clearly what do you mean: “homogenized organ pools of each sample”
|
|
Response 2: Thank you for your comment. As reported in the Materials and Methods section (sample collection subsection), for each animal included in this study, multiple organs were sampled (lines 135–136). Following collection, all organs from the same individual were placed in a single Petri dish (one dish per sampled animal). This was done because virological testing is performed on a pooled sample of all the organs collected from each individual. These pooled tissues are mechanically homogenized prior to analysis.
|
|
Comment 3: Line 134 Possibly the subsection title could be changed for “PCR detection” (for example) as you did not perform the Molecular analysis
|
|
Response 3: Thank you for pointing this out. We agree with this comment and, accordingly, we changed the titled of the Materials and method subsection number 2.3.
|
|
Quality of English Language |
|
We thank you for your feedback regarding the quality of the English language in our manuscript. In order to improve it, we have decided to make use of the language editing service provided by MDPI. We will request this service after the revision process is complete, as the exact word count of the manuscript is required. |

Reviewer 2 Report
Comments and Suggestions for Authors
Virological Passive Surveillance of Avian Influenza and Arboviruses in Wild Birds: A Two-Year Study (2023-2024) in Lombardy, Italy
The paper describes the detection of AIVs, WNV, and USUV in wild birds from Lombardy during 2023–2024.
Corrections/suggestions
- Line 31: The lineages of WNV is missing.
- Line 32: Which co-infections? WNV/USUV, WNV/AI, USUV/AI? This should be clarified.
- Line 51: The name of the virus genus is inconsistent between the abstract and introduction sections: it is referred to as "genus Influenzavirus A" in the introduction and "Orthomyxovirus genus" in the abstract. This should be corrected in the abstract.
- Line 59: Replace symptoms by clinical signs (suggestion)
- Line 63-64: The names of the bird’s orders should be written in italics.
- Line 21/line 27: It is necessary to standardize the writing. -20 ºC or -20ºC (without space).
- Line 142/143: Correct the order of the bibliographic references. Reference 35 does not pertain to the influenza virus!
- Line 144: ZSVe or IZSLER (line 136)?
- Line 149: In the list of authors, you do not have any colleagues from the IZSAM Reference Center in Teramo. Is that correct?
- Table 1: Family name should be in italic.
- Line 210: It is not clear in the methods section how you identified the AIV clade. Was it through nucleotide sequencing? If so, the methodology applied should be described in the materials and methods section.
- Figure 2 and Table1 should be placed in the appendix section (supplementary files) to provide a better reading of the work.
- Line 254: “…this AIV was confirmed as the causative agent of numerous outbreaks affecting both domestic and wild birds across several Euro”. References should be added.
- Line 257: references should be added at the endo of the sentence.
- Line 334: Replace All by Most, at the beginning of the sentence.
- Line 368: Replace disease by virus circulation.
- Line 393: Reference is missing.
- Line 140: Remove space between words.
Author Response
|
Response to Reviewer 2 Comments
|
|
Thank you very much for taking the time to review this manuscript. Please find below our detailed responses to each of your comments, with the corresponding revisions clearly highlighted in the revised manuscript using track changes. Due to the modifications and additions made to the text, the line numbers in the revised version differ from those in the original manuscript. The line references provided in our responses correspond to the revised file.
|
|
Correction/suggestions |
|
Comment 1: Line 31: The lineages of WNV is missing.
|
|
Response 1: Thank you for pointing this out. We added this information in the text (line 32).
|
|
Comment 2: Line 32: Which co-infections? WNV/USUV, WNV/AI, USUV/AI? This should be clarified.
|
|
Response 2: Thank you for your comment. We clarified this point in the text (line 33).
|
|
Comment 3: The name of the virus genus is inconsistent between the abstract and introduction sections: it is referred to as "genus Influenzavirus A" in the introduction and "Orthomyxovirus genus" in the abstract. This should be corrected in the abstract.
|
|
Response 3: Thank you for your comment. We have corrected the information regarding the viral genus and family in the Abstract (line 19), the Introduction (line 52), and the Results section (subsection 3.1, Prevalence of AIVs, line 218), ensuring consistency across the manuscript. These corrections were made in accordance with the classifications provided by the International Committee on Taxonomy of Viruses (ICTV). · |
|
Comment 4: Line 59: Replace symptoms by clinical signs (suggestion) · |
|
Response 4: Thank you for your comment. We agree with your suggestion and replaced “symptoms” with “clinical signs” in line 61. · |
|
Comment 5: Line 63-64: The names of the bird’s orders should be written in italics. · |
|
Response 5: Thank you for your comment. Taxonomic orders mentioned in lines 64-65 have been corrected and formatted in italics.
|
|
Comment 6: Line 21/line 27: It is necessary to standardize the writing. -20 ºC or -20ºC (without space). · |
|
Response 6: Thank you for your comment. We standardize the writing in the text, adding the space in line 140. · |
|
Comment 7: Line 142/143: Correct the order of the bibliographic references. Reference 35 does not pertain to the influenza virus! · |
|
Response 7: Thank you very much for your comment. We corrected the bibliographic references related to AIV detection protocols (lines 160). · |
|
Comment 8: Line 144: IZSVe or IZSLER (line 163)? · |
|
Response 8: Thank you for pointing this out. After confirming the AIV positivity of the sample and identifying the presence of an H5 or H7 genotype at the IZSLER, the samples were sent to the Istituto Zooprofilattico Sperimentale delle Venezie (IZSVe), according to national legislations and indication given by the Italian Ministry of Health. In fact, as National Laboratory reference for Ai and ND, IZSVe is responsible charged for confirming diagnoses made by other laboratories and for virus characterization.
In order to avoid misunderstanding or doubts, we added the full name of the institution before the acronym (lines 159-162).
|
|
Comment 9: Line 149: In the list of authors, you do not have any colleagues from the IZSAM Reference Center in Teramo. Is that correct? · |
|
Response 9: Thank you for the comment. Yes, that is correct: all virological analyses aimed at the identification of WNV and USUV, including lineage determination, were performed at IZSLER by our co-author colleagues, as it is the competent institute for the region in which the research was conducted and considering that they have acquired a quite long expertise and knowledge on this disease that was firstly reported in Italy in the same area more than 15 years ago. At the National Reference Centre for Exotic Diseases (CESME) of the Istituto Zooprofilattico Sperimentale dell’Abruzzo e del Molise “G. Caporale,” only the confirmation of our results was carried out, according to national legislations and indication given by the Italian Ministry of Health.
At lines 172-174, we added the full name of the institution.
|
|
Comment 10: Table 1: Family name should be in italic.
|
|
Response 10: Thank you for your comment. We have corrected the table by formatting the wild bird families in italics. · |
|
Comment 11: Line 210: It is not clear in the methods section how you identified the AIV clade. Was it through nucleotide sequencing? If so, the methodology applied should be described in the materials and methods section. · |
|
Response 11: Thank you for pointing this out. It is important to note that, in Italy, all samples testing positive for avian influenza viruses H5 and H7 must be sent, according to national legislation and following the indications of the Ministry of Health, to the National Reference Laboratory located at the IZSVe in Padua. At this institution, H5 and H7 influenza viruses undergo specific analyses (Reverse Transcriptase real-time PCR essays) to determine their pathotype and their clade, as described in protocols PDPVIR1005 and PDPVIR125, as well as neuraminidase subtyping (protocol code PDPVIR1004). These procedures are routinely followed by genotypic and phylogenetic analyses, conducted after whole-genome sequencing of the viral genome using Next-Generation Sequencing (NGS) technology, but we did not consider this information in our research. To clarify these steps, we have added a paragraph to the “Materials and Methods” section (subsection Detection of Viral Genomes, lines 162–167), and included bibliographic references.
|
|
Comment 12: Figure 2 and Table1 should be placed in the appendix section (supplementary files) to provide a better reading of the work. · |
|
Response 12: Thank you for your comment. We have removed Table 1 from the main manuscript and included it in the Supplementary Material (Table S1), updating the corresponding reference in the text (lines 178-179).
With regard to Figure 2, and in light of the comments provided in the text (lines 190–200), we would prefer to retain it in the main manuscript at this stage. However, if the Editor will deem it more appropriate to relocate Figure 2 to the Supplementary Material, we will be pleased to make this adjustment.
|
|
Comment 13: Line 254: “…this AIV was confirmed as the causative agent of numerous outbreaks affecting both domestic and wild birds across several Euro”. References should be added. · |
|
Response 13: Thank you for your comment. We added the references at the end of the line (line 283). · |
|
Comment 14: Line 257: references should be added at the endo of the sentence. · |
|
Response 14: Thank you for your comment. We added the refences at the end of the line 285.
|
|
Comment 15: Line 334: Replace All by Most, at the beginning of the sentence. · |
|
Response 15: Thank you for you comment. We replaced it as you suggest (line 362). · |
|
Comment 16: Line 368: Replace disease by virus circulation. · |
|
Response 16: Thank you for your comment. We modified the text (line 396). · |
|
Comment 17: Line 393: Reference is missing. · |
|
Response 17: Thank you for your comment. We added the refences at the end of the line 421. · |
|
Comment 18: Line 140: Remove space between words. · |
|
Response 18: Thank you for your comment. We removed the space between “Europe” and “resulting” (line 439). · |
|
Quality of English Language |
|
· We thank you for your feedback regarding the quality of the English language in our manuscript. In order to improve it, we have decided to make use of the language editing service provided by MDPI. We will request this service after the revision process is complete, as the exact word count of the manuscript is required. |
